# Active suppression of a leaf meristem orchestrates determinate leaf growth

**John Paul Alvarez[1]\*, Chihiro Furumizu[1], Idan Efroni[2], Yuval Eshed[3]\*, John L Bowman[1,4]\***

[1]School of Biological Sciences, Monash University, Melbourne, Australia; [2]The Robert H. Smith Institute of Plant Sciences and Genetics in Agriculture, The Hebrew University of Jerusalem, Rehovot, Israel; [3]Department of Plant and environmental Sciences, Weizmann Institute of Science, Rehovot, Israel; [4]Department of Plant Biology, University of California, Davis, Davis, California, United States

**Abstract** Leaves are flat determinate organs derived from indeterminate shoot apical meristems. The presence of a specific leaf meristem is debated, as anatomical features typical of meristems are not present in leaves. Here we demonstrate that multiple NGATHA (NGA) and CINCINNATA-class-TCP (CIN-TCP) transcription factors act redundantly, shortly after leaf initiation, to gradually restrict the activity of a leaf meristem in *Arabidopsis thaliana* to marginal and basal domains, and that their absence confers persistent marginal growth to leaves, cotyledons and floral organs. Following primordia initiation, the restriction of the broadly acting leaf meristem to the margins is mediated by the juxtaposition of adaxial and abaxial domains and maintained by WOX homeobox transcription factors, whereas other marginal elaboration genes are dispensable for its maintenance. This genetic framework parallels the morphogenetic program of shoot apical meristems and may represent a relic of an ancestral shoot system from which seed plant leaves evolved.

**\*For correspondence:** john.alvarez@monash.edu (JPA); yuval.eshed@weizmann.ac.il (YE); John.Bowman@monash.edu (JLB)

**Competing interests:** The authors declare that no competing interests exist.

## Introduction

Traditionally, plant organs are divided into organs with indeterminate growth such as shoots, roots and vascular cambia, whose growth is maintained by meristems, groups of pluripotent cells, and organs with determinate growth such as leaves or floral organs. Fossil evidence indicates that seed plant leaves evolved from ancestral shoot systems, and further, the dichotomous morphology of early seed plant leaves suggests growth via a persistent apical meristem (reviewed in [*Kenrick and Crane, 1997*; *Floyd and Bowman, 2010*]). However, as anatomical features typical of apical or vascular meristems are not present in leaves, whether developing leaves grow from a localized meristem has been debated for nearly a century (*Foster, 1936*; *Hagemann and Gleissberg, 1996*).

In one of the first detailed examinations of development at the plant shoot apex Caspar Wolff described the leaf lamina arising from the margins of *Brassica 'capitata'* (cabbage) leaves (*Wolff, 1759*). Subsequently, Avery suggested that early lamina growth of *Nicotiana tabaccum* was initiated by a row of subepidermal initial cells located at the upper-lower (adaxial-abaxial) leaf boundary that he termed the 'marginal meristem' (*Avery, 1933*). However, it had already been noted that later protracted growth in leaves occurs in tissues that are not marginal, but rather within the developing lamina in a region described as a 'plate meristem' (*Schüepp, 1918*, *1926*). Thus, early views of leaf development were perceived to consist of two growth phases (*Foster, 1936*). An early ephemeral phase of cell divisions without cell expansion produces the characteristic 6–10 cell layers of the leaf thickness via submarginal periclinal cell divisions and epidermal anticlinal divisions. This is followed by a later prolonged growth phase where the bulk of two-dimensional lamina

growth is produced via a plate meristem in which cell divisions are predominantly anticlinal. Analyses of leaf development in the middle of the 20th century sought to identify patterns of submarginal cell divisions to identify initial cells, but the patterns of cell division were highly variable between species casting doubt on the presence of specific initials (*Foster, 1936*).

More recently, examination of mitotic indices during leaf development revealed that a higher rate of cell division is observed in submarginal (i.e. plate meristem) regions of the leaf as compared to the margins (*Maksymowych and Erickson, 1960*; *Fuchs, 1966*; *Thomasson, 1970*; *Dubuc-Lebreux and Sattler, 1981*; *Jéune, 1981*). Furthermore, sector analysis of leaf development in several eudicot species, including *N. tabacum*, revealed that most clonal sectors were located between the midrib and the margin, with only a minority extending all the way to the margin (*Dulieu, 1968*; *Poethig and Sussex, 1985*; *Dolan and Poethig, 1998*), indicating that leaves do not grow from the margins sensu stricto, and calling into question the concept of the leaf marginal meristem. However, noting the overall lack of organized cell division patterns in plants, Hagemann and Gleissberg argued that the defining features of meristems are their organogenetic potential and cytohistological state rather than specific cell division patterns. Thus, in their view the marginal meristem (or 'blastozone', as they refer to it) is responsible for primary morphogenetic events, e.g. lamina initiation, but is used up early in leaf development, with most lamina expansion occurring during a later leaf differentiation phase (*Hagemann and Gleissberg, 1996*). Although this is a compelling model, evidence for this interpretation has been mainly observational and circumstantial.

Here we show that removal of multiple growth suppressing transcriptional factors results in indeterminate growth of the margins of all lateral organs, coupled with sustained organogenesis and activity of gene modules shared amongst other plant meristems. Our finding supports the presence of a specific leaf meristem, and conforms to the view stemming from the fossil record that recruitment of suppressors of meristematic activity was critical in seed plant leaf evolution and development.

## Results

In *Arabidopsis*, leaf morphogenesis is initiated at the flanks of the shoot apical meristem (SAM) where leaf primordia develop as flattened lamina with defined abaxial, adaxial and marginal cell types (*Tsukaya, 2013*). Lamina development requires the juxtaposition of abaxial/adaxial polarity factors, including adaxial class III HD-Zip and abaxial KANADI transcription factors. These lie on either side of a narrow middle domain expressing the *WUSCHEL RELATED HOMEOBOX* (*WOX*) genes, *PRESSED FLOWER* (*PRS*) and *WOX1*, and together promote organ growth and differentiation (*Nakata et al., 2012*; *Wang et al., 2011*; *Eshed et al., 2004*). In *Arabidopsis* leaf development, expression of growth genes rapidly diminishes distally but can persist proximally (*Donnelly et al., 1999*; *Nath et al., 2003*). This proximo-distal differentiation gradient is regulated by CIN-TCP transcription factors (*Nath et al., 2003*). A reduction of five CIN-TCPs targeted by the endogenous microRNA, miR319a (also known as miR-JAW) results in delayed basipetal progression of a mitotic arrest front and increased cell proliferation particularly at leaf margins, producing crinkly and serrated leaves (*Efroni et al., 2008*; *Ori et al., 2007*; *Palatnik et al., 2003*). Increased distal leaf growth and serrations are also observed when the activities of the four NGA transcription factors are reduced (*Figure 1—figure supplement 1*) (*Trigueros et al., 2009*; *Alvarez et al., 2009*). The NGAs and CIN-TCPs are co-expressed at many stages of leaf development, exemplified by the distal expression of *TCP3*, *TCP4*, *NGA1* and *NGA4* in young leaves (*Figure 1—figure supplement 2*) and in contrast to the reported expression of *miR319* at the leaf base (*Obayashi et al., 2009*; *Nag et al., 2009*). This, together with similarities in their loss-of-function phenotypes, suggests shared roles in leaf development. To investigate functional redundancy, we introduced a constitutive expression construct of miR319a (*35S:miR319*) targeting the five *CIN-TCP* genes (*Palatnik et al., 2003*) into a quadruple *NGA* mutant (*nga1,2,3,4*) that lacks NGA activities.

Strikingly, simultaneous reduction in expression of these nine genes resulted in continuous *de novo* formation of tissue at the margins of all lateral organs including cotyledons, leaves and floral organs (*Figure 1A–B*, *Figure 1—figure supplement 3*). Indistinguishable phenotypes were observed in plants constitutively expressing both *miR319a* and the previously characterized artificial miRNA *amiR-NGA* (*Alvarez et al., 2009*), facilitating easier and more extensive characterization of the indeterminate growth phenotype. *35S:miR319a/35S:amiR-NGA* plants grow more slowly, are later

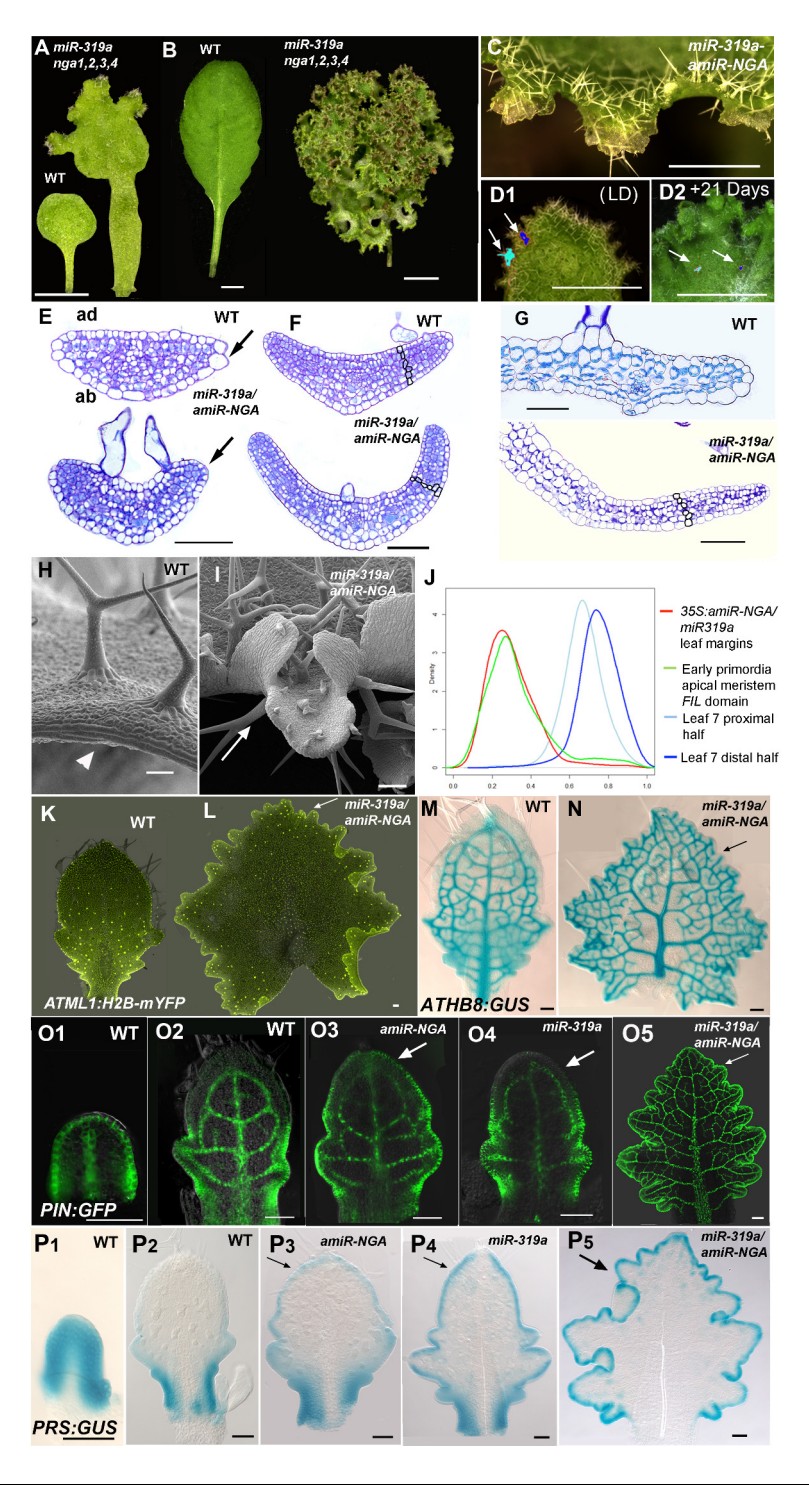

**Figure 1.** Reduction in NGA and CIN-TCP activities renders organ margins indeterminate. (**A**, **B**) Overview of wild-type (WT) and *35S:miR319a nga1,2,3,4* (*miR319a nga1,2,3,4*) cotyledons (**A**) and leaves (**B**). (**C**-**I**) Close ups of leaf margins of WT and *35S:miR319a-amiR-NGA* plants. (**D1**-**D2**) Third leaf of *35S:miR319a/35S:amiR-NGA* plant marked with nail polish (arrows; light blue and dark blue), demonstrating the ongoing growth from the margins along 21 days. (**E**) Transverse sections through the distal end of developing wild-type and *35S:miR319a/35S:amiR-NGA* leaves showing large versus small marginal cells (arrows). (**F**) Developing wild-type and *35S:miR319a/35S:amiR-NGA* leaf primordia exhibit a similar six-cell-layered blade anatomy (outlined). (**G**) The margin of older wild-type margins have large, differentiated cells, whereas the *35S:miR319a/35S:amiR-NGA* leaf margin retains the primordial blade structure. (**H**) The marginal cells of wild-type are elongated (arrowhead), while small isodiametric cells and initiating trichomes are found at *35S:miR319a/35S:amiR-NGA* margins (**I**; arrow). (**J**) Transcriptome-based differentiation-score distributions of dissected *35S:*

*Figure 1 continued on next page*

*Figure 1 continued*

*miR319a/35S:amiR-NGA* leaf margins, sorted primordia cells expressing *FIL* gene, and proximal or distal halves of seventh wild-type leaves (see Materials and methods for details). (K-L) Expression of *ATML1:H2B-mYFP* (yellow), *ATHB8:GUS* (M, N; blue), *PIN1:PIN1-GFP* (O; green), and *PRS:GUS* (P; blue) in developing leaves of indicated genotypes and a young wild-type leaf primordium (O1, P1). Note the distal exclusion of marker expression in slightly older wild-type leaves (O2, P2) while arrows indicate persisting expression along the distal margin with reduced NGA and CIN-TCP activities. ad, adaxial leaf side; ab, abaxial leaf side. Scale bars: A, D1-D2, 2 mm; B, 5 mm; C, 1 mm; 50 µm in other panels.

The following source data and figure supplements are available for figure 1:

**Source data 1.** Mean size of the leaf in wild-type and 35S:amiR-NGA plants, corresponding to the data shown in *Figure 1—figure supplement 1C*.
**Source data 2.** Mean size of the palisade mesophyll cells in wild-type and 35S:amiR-NGA plants, corresponding to the data discussed in the legend to *Figure 1—figure supplement 1D and E*.
**Source data 3.** CYCB1;1:GUS expression in distal wild-type and 35S:miR-NGA, correspnding to the data shown in *Figure 1—figure supplement 1F–I*.
**Source data 4.** Effects on expression of different CIN-TCP and NGATHA family members and possible off targets in amiR-NGA and miR319a overexpressing plants- *Figure 1—figure supplement 3*.
**Source data 5.** Differences in flowering time among wild-type, 35S:amiR-NGA, 35S:miR319a and 35S:amiR-NGA/35S:miR319a plants, corresponding to the data shown in *Figure 1—figure supplement 4D*.
**Figure supplement 1.** Altered growth in leaves with reduced NGATHA gene activity.
**Figure supplement 2.** *NGA1:GUS, NGA4:GUS, TCP4:GUS* and *TCP3:GUS* expression in leaves.
**Figure supplement 3.** Ongoing marginal growth in leaves and floral organs with reduced CIN-TCP and NGATHA gene activities.
**Figure supplement 4.** Plant growth and flowering time in plants with reduced CIN-TCP and NGATHA gene activities.
**Figure supplement 5.** SEM of leaf margin cells with reduced CIN-TCP and NGATHA gene activities.
**Figure supplement 6.** Patterns of proximal to distal leaf margin cell expansion.
**Figure supplement 7.** Distribution of markers in leaves with reduced CIN-TCP and NGATHA gene activities.
**Figure supplement 8.** Transverse sections of leaves with reduced NGATHA and CIN-TCP activities.
**Figure supplement 9.** Changes in PIN1-GFP expression when CIN-TCPs and NGATHA gene activities are reduced.
**Figure supplement 10.** *ATHB8:GUS* expression in leaves with reduced CIN-TCPs and NGATHA gene activities.
**Figure supplement 11.** *PRS:GUS* expression is maintained longer at the leaf margins when CIN-TCP and NGATHA gene activities are reduced.
**Figure supplement 12.** *WOX1:GUS* expression is maintained longer at the leaf margins when CIN-TCP and NGATHA gene activities are reduced.
**Figure supplement 13.** Reduced NGATHA and CIN-TCP gene activities in the *PRESSED FLOWER (PRS)* domain alters leaf marginal growth.
**Figure supplement 14.** Ongoing marginal growth in cotyledons with reduced CIN-TCP and NGATHA gene activities.
**Figure supplement 15.** Morphological and marker analyses in cotyledons with reduced CIN-TCP and NGATHA gene activities.

flowering, and their leaf margins harbor proliferative cell populations unlike those of *35S:amiR-NGA* and *35S:miR319a* singly transgenic plants (*Figure 1C–G*, *Figure 1—figure supplement 3–8*). Application of stain to *35S:miR319a/35S:amiR-NGA* leaf margins indicates continued proliferation at the leaf margin, with the marker displaced sub-marginally over time (*Figure 1D1–D2*, *Figure 1—figure supplement 3*). In *35S:miR319a nga1,2,3,4* or *35S:miR319a/35S:amiR-NGA* plants, the entirety of

the older leaf margin consists of small densely packed cells lacking chlorophyll, rather than the large, elongate cells characteristic of wild-type leaf margins (*Figure 1C–I*, *Figure 1—figure supplement 5–8*). Sections of leaf primordia and differentiating leaves suggest that the six-cell-layered blade organization of young wild-type leaf primordia is maintained at *35S:miR319a/35S:amiR-NGA* leaf margins ([*Nakata et al., 2012*]; *Figure 1E–G*, *Figure 1—figure supplement 8*).

The digital differentiation index (DDI) assesses relative leaf maturity from global gene expression profiles (*Efroni et al., 2008*). The index of dissected, older *35S:miR319a/35S:amiR-NGA* leaf margins clearly matches that of initiating primordia (*Figure 1J*). This result is further supported by the expression of markers that highlight continued cell division, distinguishing epidermal nuclei (*ATML1:H2B-mYFP*), epidermal plasma membrane (*ATML1:mCitrine-RCI2A*), general cell division (*CYCB1;1:GFP*), initiating trichomes (highlighted by *GL2:GFP*) and stomatal lineage proliferation (*TMM:GUS-GFP*) (*Figure 1K,L*, *Figure 1—figure supplement 7*), which demonstrate ongoing leaf-primordium-like activity at the leaf margins. In initiating wild-type leaves, auxin flux, marked by PIN-FORMED1 (PIN1) expression, converges at the distal tip and at serrations, where it inwardly canalizes leaf vascular development, before becoming restricted to proximal margins of older leaves (*Bilsborough et al., 2011*; *Scarpella et al., 2006*). Compared to wild-type leaves, in both *35S:amiR-NGA* and *35S:miR319a* individual knockdown leaves auxin flux persists longer at distal leaf margins. Strikingly, in the *35S:miR319a/35S:amiR-NGA* combined knockdown leaves, auxin flux continues around the entire leaf margin (*Figure 1O*, *Figure 1—figure supplement 9*). Auxin canalization and ongoing *de novo* vasculature morphogenesis at these margins is marked by expression of the pro-vascular makers *ATHB8* and *MONOPTEROS* (*MP*) (*Figure 1M–N*, *Figure 1—figure supplements 7* and *10*). Paralleling marginal auxin flux, the organ marginal markers *PRS and WOX1* are transiently expressed in initiating wild-type leaves before becoming proximally restricted. When NGA or CIN-TCP activities are reduced, *PRS and WOX1* distal expression persists in older leaves whereas in the combined loss in *35S:miR319a/35S:amiR-NGA* leaves *PRS* and *WOX1* expression occurs in an uninterrupted marginal band, again suggesting that these margins retain meristematic properties equivalent to initiating leaf primordia (*Figure 1P*, *Figure 1—figure supplements 11* and *12*).

That expression of *miR319a-amiR-NGA* under control of the *PRS* regulatory sequences results in indeterminate margins confirms that marginal loss of NGA and CIN-TCP activity is sufficient to allow the maintenance of these meristematic characteristics (*Figure 1—figure supplement 13*). Notably the lamina away from the margins of *PRS>>miR319a-amiR-NGA* is thinner and more wild-type in appearance than that of *35S:miR319a/35S:amiR-NGA* leaves suggesting that the broader, non-marginal expression of the *NGA*s and *CIN-TCP*s may reflect an activity in regulating cell expansion that remains functional in *PRS>>miR319a-amiR-NGA* leaves (*Figure 1—figure supplement 8* and *Figure 1—figure supplement 13*).

The extended maintenance of primordium identity was also observed in the cotyledons of *35S:miR319a nga1,2,3,4* or *35S:miR319a/35S:amiR-NGA* plants, which continuously produce tissue with leaf characteristics including stellate trichome formation (*Figure 1A*, *Figure 1—figure supplement 14*). Changes in the expression pattern of the cell division marker *CYCB1;1:GFP* are apparent in the distal embryonic cotyledons while the respective expression of *ATML1:H2B-mYFP* and *MONOPTEROS* demonstrates the absence of the normal, marginal cell differentiation program and ectopic production of provascular strands implying an active marginal meristem similar to that observed in leaves (*Figure 1—figure supplement 15*). Notably there was no evidence for impaired dormancy of *35S:miR319a nga1,2,3,4* or *35S:miR319a/35S:amiR-NGA* seed suggesting that the seed-based program of imposed dormancy was as effective on this cotyledon marginal meristem as on the embryonic shoot and root meristems. After germination, cotyledons of *35S:miR319a nga1,2,3,4* or *35S:miR319a/35S:amiR-NGA* seedlings continue growth and express growth markers unlike wild-type (*Figure 1—figure supplements 14* and *15*). The floral organs of NGA and CIN-TCP compromised plants also exhibit prolonged marginal growth (*Figure 1—figure supplement 3*). Hence NGA and CIN-TCP redundantly suppress marginal growth in all aerial lateral organs.

Ongoing growth from the organ margin may be a consequence of ectopic activation of a SAM program. We surveyed the expression of genes that are expressed in the SAM but not in leaves of *Arabidopsis*, and no evidence was found for the expression of meristem genes including *SHOOT MERISTEMLESS* (*STM*), *WUSCHEL* (*WUS*) and *CLAVATA1/3* (*CLV1/3*) in indeterminate leaf margins of *35S:miR319a/35S:amiR-NGA* plants (*Figure 2A–E*). In agreement, the *35S:miR319a-amiR-NGA* transgene conferred indeterminate growth of cotyledon and/or leaf margins in *stm-11 knat6-1 bp-9*

triple and *wus-1* single mutants where SAM activity is respectively lost or disrupted (*Figure 2F–L*). Thus, continued marginal growth in *35S:miR319a-amiR-NGA* double knockdown leaves is not a consequence of secondarily acquiring characteristics of the indeterminate SAM, as for example in YABBY-compromised mutants (*Sarojam et al., 2010*).

The NAC transcription factors *CUP-SHAPED COTYLEDON2* (*CUC2*) and *CUC3* are regulators of leaf margin shape in *Arabidopsis* and other angiosperm species, and ectopic activation of *CUC* genes promotes adventitious shoot formation (*Blein et al., 2008*; *Aichinger et al., 2012*; *Hibara et al., 2003*), suggesting that deregulation of *CUC* genes may account for the indeterminate growth phenotype. However, we found that the cotyledon and/or leaf margins continue to grow in

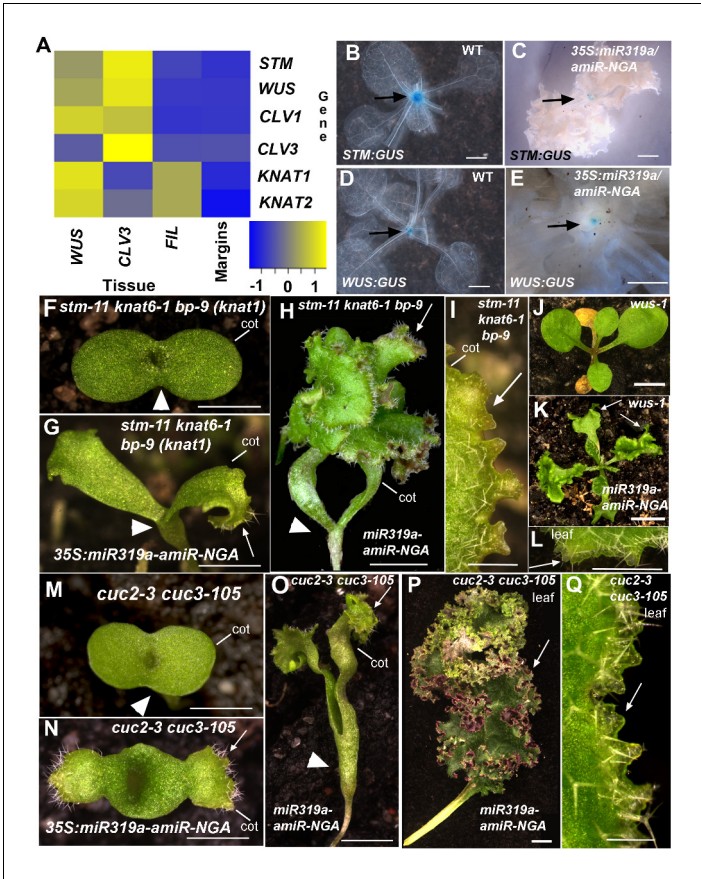

**Figure 2.** Regulators of SAM maintenance and leaf margin elaboration are dispensable for maintenance of the marginal leaf meristem. (**A**) Relative mRNA expression levels of meristem regulators in cells collected from apices and in indeterminate leaf margins. Levels of three class 1 KNOX genes, *STM*, *KNAT1* and *KNAT2*, as well as *WUS* and *CLV1/3* are shown. Levels were determined in sorted cells expressing *WUS* or *CLV3* (meristem-expressed) or *FIL* (expressed in developing organs) (see Materials and methods for details) and compared with those in *35S: miR319a/35S:amiR-NGA* indeterminate leaf margins (labeled as Margins). Heatmap color represents the row Z-score. (**B**, **C**) *STM:GUS* and (**D**, **E**) *WUS:GUS* expression (blue) is confined to the vegetative shoot meristem (arrows) both in wild-type and *35S:miR319a/35S:amiR-NGA* plants. (**F–I**) *stm-11 knat6-1 bp-9*, (**J–L**) *wus-1*, and (**M-Q**) *cuc2-3 cuc3-105* seedlings have a disrupted apical meristem. Arrowheads denote fused cotyledons. In the presence of the *35S:miR319a-amiR-NGA* transgene, cotyledons (**G, H, I, K, N, O**) and leaves (**K, L, P, Q**) grow indeterminately (arrows) in these mutants. Close-ups of the indeterminate margin are shown in **I, L, Q**. Leaves in **P** and **Q** are produced from a SAM that grows through fused *cuc2-3 cuc3-105* cotyledons such as those depicted in **O**. cot, cotyledon. Scale bars: **B-E**, **J**, **K**, 5 mm; **F-H**, **M-O**, 2 mm; **I, L, Q**, 0.5 mm; **P**, 1 cm.

The following figure supplement is available for figure 2:

**Figure supplement 1.** CUC activities are dispensable for indeterminate leaf margin growth.

*cuc2 cuc3* and *cuc1 cuc2* mutants expressing the *35S:miR319a-amiR-NGA* transgene. This indicates that continued margin growth is independent of CUC-mediated marginal elaboration (*Figure 2M–Q*, *Figure 2—figure supplement 1*).

Since lamina growth is an outcome of an interaction between adaxial and abaxial factors and involves the marginal leaf WOX genes (*Nakata et al., 2012*; *Eshed et al., 2004*), the role of polarity factors and WOX genes in maintaining continued marginal growth was investigated. The respective adaxial, marginal and abaxial genes *PHABULOSA* (*PHB*), *PRS* and *KANADI1* (*KAN1*) are expressed in young, wild-type leaf primordia before diminishing in a basipetal fashion (*Figure 3A,C,E,G–H*, *Figure 3—figure supplement 1*). At the margins of *35S:miR319a/35S:amiR-NGA* leaves, *PHB*, *PRS* and *KAN1* gene expression continues indefinitely, with spatial relationships maintained, implying that in older leaves with reduced CIN-TCP and NGA activities, the collective interplay among these genes is sustained as established in initiating wild-type leaf primordia (*Figure 3A–H*, *Figure 1—figure supplement 11*, *Figure 3—figure supplement 1*). To test whether adaxial/abaxial tissue polarity and associated WOX activities are required for marginal leaf growth we examined the effects of mutations in these genes on indeterminate marginal growth. Semi-dominant *PHB* alleles produce two leaf types on the same plant: partially radialized leaves with distal lamina and completely radialized (adaxialized) leaves (*Figure 3I*). In *35S:miR319a/35S:amiR-NGA phb-1d/+* plants, leaves with distal lamina exhibited ectopic marginal growth while radialized leaves did not, demonstrating that ongoing marginal growth first requires the juxtaposition of polarity factors (*Figure 3J*, *Figure 3—figure supplement 2*). *PRS* and *WOX1* redundantly promote growth as an output of the abaxial/adaxial polarity program (*Nakata et al., 2012*). The combined loss of NGA and CIN-TCP activities in *35S:miR319a/35S:amiR-NGA* plants results in both *PRS* and *WOX1* expression occurring as an uninterrupted marginal band in older leaves (*Figure 1P*, *Figure 1—figure supplements 11* and *12*). Notably, *prs wox1* double mutants suppressed the indeterminate marginal growth in *35S:miR319a/35S:amiR-NGA* plants (*Figure 3K–L*, *Figure 3—figure supplement 3*). Hence the ongoing leaf margin growth is dependent on both the polarity program and the leaf-specific *WOX* genes.

To further characterize the relationships between the different leaf domains, we investigated weak polarity mutant backgrounds where ectopic sites of adaxial/abaxial juxtaposition lead to outgrowths, which have marginal identity, from the leaf lamina (*Nakata et al., 2012*; *Wang et al., 2011*; *Eshed et al., 2004*). The abaxial surfaces of developing *kan1 kan2* mutant leaves exhibit ectopic expression of *PIN1*, *PRS* and *NGA1* (*Figure 3M–P*). Reducing both NGA and CIN-TCP activity in the *kan1 kan2* background results in a striking proliferation of leaf tissue from the abaxial surface (*Figure 3Q*, *Figure 3—figure supplement 2*). Similarly, reducing NGA and CIN-TCP activities in mutants of the adaxial factor *ASYMMETRIC LEAVES2* (*AS2*), where patches of ectopic, adaxial *PRS* expression are observed, resulted in adaxial lamina proliferation (*Figure 3R–U*, *Figure 3—figure supplement 2*). A shift in the marginal program with corresponding lamina outgrowths can also be achieved through direct manipulation of *WOX1* expression, such as ectopic abaxial expression of *WOX1* in *FIL:WOX1* plants (*Figure 4A–C*) (*Nakata et al., 2012*). Here, as in *kan1 kan2* mutant leaves, we detected *PIN1* and *NGA1* expression in the abaxial outgrowths.

The indeterminate cell proliferation and patterning of the leaf margin in *35S:miR319a/35S:amiR-NGA* plants suggests it is self-organizing, a property of meristems, consistent with results demonstrating positive and negative feedbacks between *PRS/WOX1* and adaxial/abaxial polarity factors (*Nakata and Okada, 2012*). The lack of marginal growth in radialized *phb-1d/+* organs and its ectopic placement at discrete positions of the lamina when the adaxial/abaxial patterning is compromised argues for a major role of the polarity factors in marginal positioning of a leaf meristem that requires the intervening activity of WOX genes. In turn, a negative feedback loop between the marginally restricted meristem and NGA/CIN-TCP activities may lead to the ephemeral nature of this meristem. In agreement, leaves of *FIL:WOX1* that are likely relieved from such feedback regulation, maintained a highly meristematic nature and failed to differentiate and expand when NGA and CIN-TCP activities were jointly reduced (*Figure 4D,E*).

## Discussion

The observation that loss of NGAs and CIN-TCPs results in indeterminate leaf margins suggests that the early wild-type leaf primordium has a meristem that acts during a brief developmental window and that is gradually restricted spatially (*Figure 4F–G*). This interpretation is consistent with classical

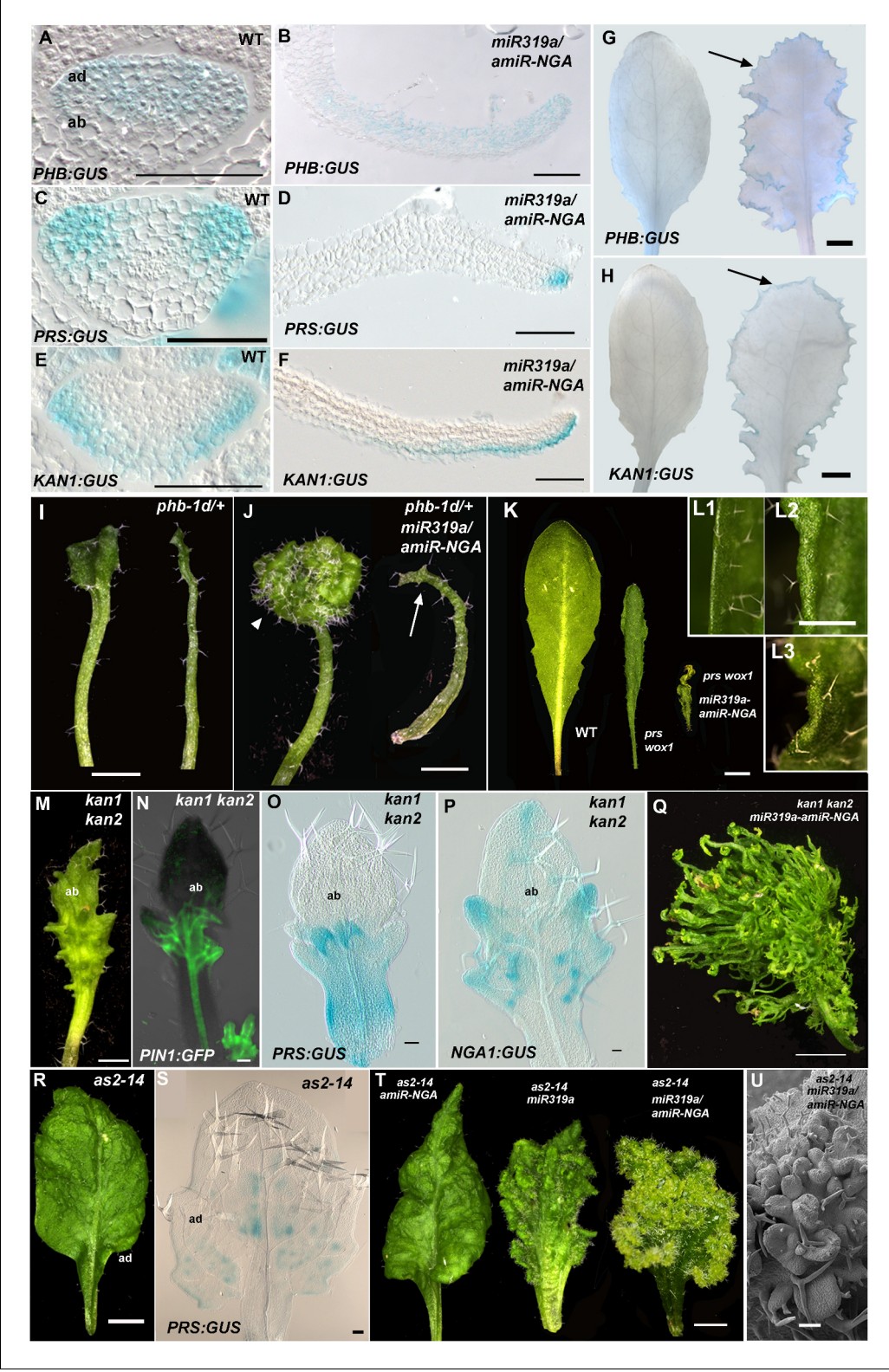

**Figure 3.** Marginal meristem activity requires juxtaposition of adaxial/abaxial polarity factors. (**A–F**) Adaxial *PHB: GUS* (**A**, **B**), central *PRS:GUS* (**C**, **D**) and abaxial *KAN1:GUS* (**E**, **F**) expression domains (blue) in transverse sections with abaxial sides facing upward. Young wild-type leaf primordia (**A**, **C**, **E**) and older *35S:miR319a/35S:amiR-NGA* leaf margins (**B**, **D**, **F**) are shown. (**G–H**) *PHB:GUS* (**G**) and *KAN1:GUS* (**H**) in whole wild-type (left) and *35S:miR319a/ 35S:amiR-NGA* (right) leaves. Arrows denote continued expression. (**I**) Completely (right) and partially (left)

*Figure 3 continued on next page*

*Figure 3 continued*

radialized leaves of *phb-1d/+* mutants. (**J**) *phb-1d /+ 35S:miR319a/35S:amiR-NGA* leaves. The distal lamina exhibits continual marginal growth (arrowhead) whereas a radialized leaf lacks such growth (arrow). (**K**) Wild-type, *prs wox1*, and *prs wox1 35S:miR319a-amiR-NGA* leaves at equivalent age. (**L**) Close-ups of differentiated leaf margins of wild-type (**L1**), *prs wox1* (**L2**) and *prs wox1 35S:miR319a-amiR-NGA* (**L3**; compare with the indeterminate leaf margin in *Figure 1C*). (**M–P**) *kan1 kan2* leaves showing abaxial outgrowths (**M**), and ectopic expression of *PIN1:PIN1-GFP* (**N**), *PRS:GUS* (**O**) and *NGA1:GUS* (**P**) associated with abaxial outgrowths. (**Q**) A *kan1 kan2 35S:miR319a-amiR-NGA* leaf showing proliferative tissue outgrowth. (**R, S**) *as2-14* leaves showing ectopic, adaxial *PRS:GUS* expression (**S**). (**T**) From left to right shown are *as2-14 35S:amiR-NGA, as2-14 35S:miR319a, as2-14 35S:miR319a/35S:amiR-NGA* leaves with increasing adaxial outgrowths. (**U**) Close-up of the adaxial surface of *as2-14 35S:miR319a/35S:amiR-NGA* leaf. ad/ab, adaxial and abaxial leaf sides. Scale bars: **G–J, M**, 2 mm; **L**, 1 mm; **K, Q, R, T**, 5 mm; 50 μm in other panels.

The following figure supplements are available for figure 3:

**Figure supplement 1.** Expression of adaxial/abaxial polarity and central WOX genes is maintained at the leaf margins when CIN-TCP and NGATHA gene activities are reduced.

**Figure supplement 2.** Adaxial/abaxial polarity factors are necessary for and spatially define the marginal meristem, which is suppressed by CIN-TCP and NGATHA gene activities.

**Figure supplement 3.** Changes in PRS/WOX1, NGATHA, or polarity factor activities affects leaf growth and morphologies.

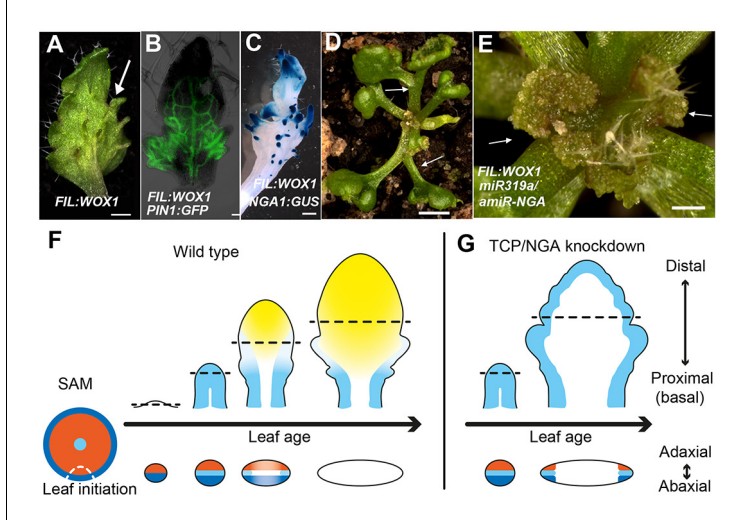

**Figure 4.** Dynamic restriction of the leaf meristem. (**A–C**) *FIL:WOX1* leaf with developing abaxial outgrowths (arrow in **A**). These outgrowths show prolonged *PIN1:PIN1-GFP* (**B**) and *NGA1:GUS* (**C**) expression. (**D, E**) In *FIL: WOX1 35S:miR319a/35S:amiR-NGA* first leaves show occasional bifurcation (arrows in **D**) and later emerging leaves are highly proliferative (arrows in **E**) in the distal domain. (**F, G**) Scheme depicting developing wild-type (**F**) and CIN-TCP/NGA compromised (**G**) leaves shown from a proximo-distal perspective (above horizontal arrows) and abaxial-adaxial perspective captured at the dashed-line (below horizontal arrows). Leaves are physically and evolutionarily derived from shoot apical meristems (SAM; left most cartoon in **F**). The SAM is radially patterned with external (blue), internal (red) and central WOX (aqua) domains. In wild-type leaf primordia (**F**) the pre-pattern at the SAM (white dash line) is converted into juxtaposed abaxial and adaxial domains directing WOX activation in an intervening domain. Feedback between these three domains stabilizes the leaf meristem, promotes lamina growth and maintains pluripotency (*Nakata and Okada, 2012*) before meristem activity is restricted to the proximal marginal domains by CIN-TCP/NGA activities (yellow), permitting prolonged growth only at the proximal region of the leaf. In leaves where CIN-TCP/NGA activities are reduced (**G**), meristem activity is maintained at all margins in a pattern reminiscent of initiating leaf primordia. Scale bars: **A, C, D**, 2 mm; **E**, 1 mm; 50 μm in **B**.

morphological and anatomical studies in which the definition of the marginal meristem was extended to include the entire meristematic leaf primordium at very early stages of leaf development (*Hagemann, 1970*). If a leaf primoridum is damaged or bifurcated at this stage, nearly complete regeneration of normal leaf morphology is possible (*Goebel, 1902*; *Figdor, 1906*; *Snow and Snow, 1941*; *Sachs, 1969*). Subsequently, as the meristematic regions become restricted to the margins or portions of the margins, damage or bifurcation of the leaf primordium results in progressively more limited regenerative capacity (*Snow and Snow, 1941*; *Sachs, 1969*; *Figdor, 1926*). *Arabidopsis* leaves are argued to possess a basal meristem that remains transiently active after leaf initiation before transitioning to petiole development — a process regulated by the *BLADE-ON-PETIOLE* (*BOP*) genes (*Hepworth et al., 2005*; *Ichihashi et al., 2011*; *Kuchen et al., 2012*; *Laux et al., 1996*). Our observations are consistent with early distal expression of CIN-TCP and NGA genes repressing the meristem distally, but the lack of early proximal expression allows marginal persistence of the leaf meristem at the leaf base, as reflected by *PRS* expression dynamics and leaf marginal cell differentiation along the proximo-distal axis (*Nakata et al., 2012*) (*Figure 1P*, *Figure 1—figure supplements 6* and *8*).

Whereas *Arabidopsis* leaves differentiate from tip to base, leaf differentiation in some other angiosperm species can proceed from base to tip (*Trécul 1853*; *Ikeuchi et al., 2013*). We thus speculate that variations in lateral organ growth within an individual and among species reflect differential maintenance of meristem activity along the marginal and proximo-distal axes. Remarkable diversity in leaf shape can arise from growth variation along the margin including leaf lobing. Lobe formation in many species relies on leaf-specific activity of class 1 KNOX (KNOX1) genes, which the simple leaves of *Arabidopsis* lack (*Piazza et al., 2010*). However, lobes can be mimicked by ectopic KNOX1 expression in *Arabidopsis* leaves. The radialized leaves of *phb-1d/+* plants, a *prs wox1* background and *NGA1* over-expression, all suppresses the KNOX1-induced lobing phenotype, indicating that an active marginal meristem is a prerequisite to respond to KNOX1 activity (*Figure 3—figure supplement 3*). Thus modulation in the marginal restriction of meristem activities can contribute to leaf shape diversity.

Cotyledons and floral organs are viewed as modified leaves. In *Arabidopsis*, lack of a basally restricted meristem may distinguish them from leaves in their response to reduced CIN-TCPs and NGA activity. In these organs, additional growth is confined to the distal region whereas in leaves the entire margin is affected (*Figure 1A–B*, *Figure 1—figure supplements 3*, *14* and *15*). The observation that leaf tissue grows from cotyledon tips suggests a brief activity of a marginal meristem in cotyledons. Prolongation of the marginal meristem activity likely uncouples growth from the embryonic cotyledon program, and therefore, cotyledons continue to grow the same way as leaves.

How can our observations of a potential continuing meristematic activity at leaf margins be reconciled with classical concepts of marginal and plate meristems in leaves and with the denial of their existence based on mitotic indices and sector analyses? Seed plant leaves evolved from ancestral shoot systems; thus, the shoot apical meristem (SAM) may provide an analogy, or perhaps homology (*Floyd and Bowman, 2010*). The seed plant SAM exhibits two distinct organizational features. Firstly, SAMs feature a tunica-corpus structure in which cell divisions in the tunica are almost exclusively anticlinal (*Schmidt, 1924*). Secondly, the seed plant SAM exhibits cytohistological zonation that is correlated with functional zonation (*Foster, 1938*). The central zone (CZ) exhibits low rates of mitoses and acts to supply cells to the peripheral zone (PZ) and rib zone (RZ) where mitotic activity is high, and organogenesis occurs (*Steeves and Sussex, 1989*). Consistent with these patterns of cell division, cell lineage analyses of the SAM reveals that the majority of sectors observed do not extend to include the SAM, but rather are presumed to originate in derivatives of the peripheral/rib zones (*Dulieu, 1969*; *Jegla and Sussex, 1989*; *Furner and Pumfrey, 1992*).

As with SAMs, leaf meristems can also be interpreted to consist of distinct organizational zones. Regions of low and high mitotic activity correspond to the classically defined 'marginal' and 'plate' meristems (*Foster, 1936*; *Avery, 1933*; *Schüepp, 1926*; *Maksymowych and Wochok, 1969*; *Maksymowych and Erickson, 1960*; *Fuchs, 1966*; *Thomasson, 1970*; *Dubuc-Lebreux and Sattler, 1981*; *Jéune, 1981*). Consistent with these mitotic indices, cell lineage analyses reveal that the majority of sectors produced in developing leaves are derived from regions internal to the margins (*Dulieu, 1968*; *Poethig and Sussex, 1985*; *Dolan and Poethig, 1998*). While marginal activity of the leaf meristem in wild-type *Arabidopsis* may be brief, we show here that when extended, cells

generated at the margins are displaced towards the center of the leaf, displaying a maturation gradient, similar to the PZ and RZ cells displaced from the CZ of the SAM.

The CZ of the SAM is characterized by the expression of a WOX gene, *WUSCHEL* (*Mayer et al., 1998*). Loss-of-function *WUS* alleles generate a functional SAM, but the CZ fails to be maintained, leading to the eventual depletion of cells in the active PZ and RZ (*Laux et al., 1996*). Similarly, the leaf meristem exhibits WOX gene expression, whose function is required for continued leaf growth, but leaves can initiate and grow for a while without marginal WOX expression (*Nakata et al., 2012*; *Vandenbussche et al., 2009*). The SAM features a tunica-corpus structure in which cell divisions in the tunica are almost exclusively anticlinal ([*Schmidt, 1924*] and others). As with the SAM, the leaf marginal domain is also organized into epidermal and sub-epidermal layers. Analysis of periclinal chimeras revealed that the epidermal layers of the leaf are clonally related, whereas the mesophyll and vascular bundles are derived from subepidermal layers ([*Foster, 1936*; *Avery, 1933*; *Baur, 1909*] and references therein). The lack of differentiation of leaf marginal cells in *35S:miR319a/35S:amiR-NGA* plants is consistent with these cells remaining meristematic.

Our results are largely consistent with classical views of leaf development — that the leaf primordium is broadly meristematic at its inception, and that meristematic potential is subsequently restricted to the marginal regions (*Foster, 1936*; *Hagemann and Gleissberg, 1996*; *Jéune, 1981*; *Sachs, 1969*). In our view, the marginal and plate meristems represent two zones of a leaf meristem, analogous, or perhaps homologous, to the central and peripheral zones of the SAM.

We suggest that the marginal restriction of the leaf meristem is in part maintained and guided by the same adaxial and abaxial factors that function in shoot and cambial meristems, and all three meristems are maintained by the activity of different *WOX* paralogs, suggesting the repeated use of a molecular module (*Figure 4F*) (*Aichinger et al., 2012*). Sharing of genetic modules implies either common descent or co-option of modules to pattern novel structures. Since seed plant leaves evolved from ancestral shoot systems, common descent is plausible. In this scenario, the leaf meristem module has been modified from an ancestral shoot meristem module to include the leaf-specific *WOX1 and PRS* paralogs (*Lin et al., 2013*; *Nardmann and Werr, 2013*) that arose in a common ancestor of seed plants. Additional regulators such as the *YABBY* genes, which are instrumental in lamina growth and restrict activity of SAM factors (*Sarojam et al., 2010*), and later acting factors limiting leaf meristem activity (i.e., CIN-TCP and NGA) were integrated into the leaf program. Growth suppressors modulating leaf meristem activity were recruited from genes of both ancient and recent origins — *CIN-TCP* genes are present in all land plants (*Navaud et al., 2007*) whereas *NGA* genes evolved recently, perhaps within seed plants (*Alvarez et al., 2009*). Thus the leaf marginal meristem genetic program may have been derived via elaboration of an ancestral shoot program, reflecting the derivation of the leaf from a modified shoot. The identification of such genetic framework provides a unification of how the entire seed plant shoot system is built from apical, vascular, cambial, and leaf meristems that are mechanistically similar. The evolution of seed plant leaves from an ancestral shoot system can be interpreted as evolving via the recruitment of regulatory mechanisms to suppress the morphogenetic potential of the leaf meristem.

## Materials and methods

### Plant material and growth conditions

For leaf analyses plants were grown under short-day conditions (10 hr light) at 20°C for 15 to 20 days.

A number of lines for genetic and image analyses were generously provided for use in this study. The *cuc2-3 cuc3-105* lines were provided by Masao Tasaka (*Hibara et al., 2006*). The *prs wox1* lines were a gift from Tom Gerats (*Vandenbussche et al., 2009*). *TMM:GUS-GFP* line was provided by Fred Sack (*Nadeau and Sack, 2002*). The *ATML1:mCitrine-RCI2A* and *ATML1:H2B-mYFP* were a gift from Adrienne Roeder (*Roeder et al., 2010*). John Celenza and Peter Doerner provided the *CycB1;1::CycB1;1-GUS* and *CycB1;1::CycB1;1-GFP* marker lines. The *PIN1:PIN1-GFP* and *DR5:GFP* were supplied by Jiří Friml (*Friml et al., 2003*). *GL2::ERGFP:NOS* was provided by Philip Benfey and Ji-Young Lee (*Lee et al., 2006*). *ATHB8:GUS* was obtained from the *Arabidopsis Biological Resource Center* (*ABRC*), Ohio State University, USA. The *MONOPTEROS/ARF5:GFP* line was gift from Dolf Weijers. The *NGA4:GUS* line is *nga4-1*, a Ds gene trap allele (SGTSET7056) (*Alvarez et al., 2009*).

Similarly *PHB:GUS* is *phb-6*, a Ds gene trap allele (SGT4606) in the first exon of *PHB* (*Hawker and Bowman, 2004*). The *BLS:STM* and *BLS* promoter, transactivation line (BLS LacI[H17]-GAL4 (LhG4)) have been previously described (*Shani et al., 2009*; *Furumizu et al., 2015*; *Lifschitz et al., 2006*). The *BLS* promoter drives gene expression in young leaf primordia but not in younger, initiating leaf primordia.

## Histology and microscopy

For tissue sections and scanning electron microscopy (SEM), samples were immersed in 2% glutaraldehyde in 0.025 M sodium phosphate buffer (pH 6.8) and vacuum infiltrated for up to one hour. For sections, specimens were then washed, dehydrated in an ethanol series, and infiltrated and embedded in LR White resin. 2 μm-thick sections were cut, dried onto slides, and stained with toluidine blue. For SEM, glutaraldehyde-fixed tissues were further fixed in 1% $OsO_4$ before dehydration through a graded ethanol series and critical point dried using liquid $CO_2$. Specimens were coated with gold in an Eiko 1B.5 sputter coater and viewed using a Hitachi s570 scanning electron microscope.

For histochemical analysis of GUS activity, samples were infiltrated with GUS staining solution [0.2% (w/v) Triton X-100, 2 mM potassium ferricyanide, 2 mM potassium ferrocyanide, and 1.9 mM 5-bromo-4-chloro-3-indolyl-β-glucuronide in 50 mM sodium phosphate buffer, pH 7.0] and incubated at 37°C.

To prepare cleared samples, tissue was fixed overnight in 9:1 (v:v) ethanol:acetic acid at room temperature. After rehydration in a graded ethanol series, samples were rinsed with water and were cleared with chloral hydrate solution [1:8:2 (v:w:v) glycerol:chloral hydrate:water], dissected, and viewed.

Fluorescence was observed using a Zeiss Axioskop2 mot plus microscope using filter set 46 for YFP (excitation BP 500/20; beam splitter FT 515; emission BP 535/30), filter set 13 for GFP (excitation BP 470/20; beam splitter FT 495; emission BP 505–530), and filter set 43 HE (excitation BP 550/25; beam splitter FT 570; emission BP 605/70) or Semrock SpOr-B-000 filter set (excitation BP 543/22; beam splitter FT 562; emission BP 586/20) for RFP. Images were collected using AxioVision software individually or as part of a Z stack that included light field and DIC (differential interference contrast) images as well. Deconvolution processing was carried out for some images.

The color of the nail polish applied to cotyledon and leaves was digitally altered to accommodate red-green colourblind viewers.

## Plasmid construction and plant transformation

Overexpression of miR319a (35S:miR319a) was carried out using a 323 bp fragment of the miR319a encoding locus including 28 bp upstream and 92 bp downstream sequences of the annotated stem-loop structure. This was cloned downstream of the 35S promoter in pART7 or the array of the lac operator (OP) sequences in a BJ36-derivative plasmid for transactivation. The *35S:amiR-NGA* and *OP:amiR-NGA* constructs used to knockdown expression of all four NGA genes have been described previously (*Alvarez et al., 2006*). To create expression constructs of the *miR319a-amiR-NGA* di-miR (two miRNAs concatemerized for co-transcription), the 323 bp, *miR319a* encoding fragment was cloned 5' of the 235 bp *amiR-NGA* gene downstream of the 35S promoter in pART7 or the array of the lac operator (OP) sequences in a BJ36-derived plasmid. Plants expressing two transgenes, *35S:amiR-NGA* and *35S:miR319a*, are labeled *miR319a/amiR-NGA* while those expressing the di-miR are labeled *miR319a-amiR-NGA*. A high proportion of plants expressing the *35S:miR319a-amiR-NGA* di-miR had a strong phenotype equivalent to F1 plants from a cross between selected, individual *35S:amiR-NGA* and *35S:miR319a* expressing lines with strong phenotypes.

To construct a GUS reporter line of *TCP4* (At3g15030), which is subject to the regulation by its endogenous miRNA, miR319, approximately 3.9 kb of the upstream sequence, which starts from the 3' end of the annotated upstream gene (At3g15020) and ends before the *TCP4* start codon, was PCR amplified and TA cloned into pCRII (Invitrogen). An approximately 1.7 kb of fragment downstream of the *TCP4* stop codon, which extends into the annotated downstream gene (At3g15040), was cloned with the miR319a target site in *TCP4* built into the forward PCR primer. The two fragments were subsequently cloned contiguously into BJ36 plasmid to create a *TCP4* promoter

cassette, and the *GUS* coding sequence was cloned between the 5' and 3' *TCP4* regulatory regions and upstream of the miR319 target site.

Similarly, to create a GUS reporter line of *TCP3* (At1g53230) subject to regulation by its endogenous miRNA, miR319, approximately 3.1 kb of the upstream sequence beginning from the 3' end of the annotated upstream gene (At1g53240) transcript and ends before the TCP3 start codon was PCR-amplified and TA cloned into pCRII (Invitrogen). An approximately 2.2 kb of fragment downstream of the *TCP3* stop codon, which extends into the annotated downstream gene (At1g53220), was cloned with the miR319a target site in *TCP3* built into the forward PCR primer. The two fragments were subsequently cloned contiguously into BJ36 plasmid to create a *TCP3* promoter cassette, and the *GUS* coding sequence was cloned between the 5' and 3' *TCP3* regulatory regions and upstream of the miR319 target site.

For the GUS marker line of *WOX1* (At3g18010), a 2.3 kb fragment upstream from the start codon and a 3.8 kb fragment downstream of the stop codon were PCR amplified and TA cloned into pCRII. The two fragments were cloned contiguously into BJ36 plasmid, and the GUS coding sequence was cloned between the upstream and downstream regulatory regions.

The *PRS/WOX3* (At2g28610) promoter GUS line was created using a PCR fragment of a 6.3 kb sequence upstream of the *PRS/WOX3* start codon. The *PRS/WOX3* promoter was cloned upstream of the *GUS* coding region in the BJ36-derivative, pRITA.

The *KANADI1:GUS* reporter line was created by cloning the *GUS* encoding DNA fragment downstream of the *KANADI1* (At5g16560) promoter that consists of a 884 bp fragment of the conserved second intron fused to a 5.3 kb fragment upstream of *KANADI1*, which has been previously described (*Efroni et al., 2008*).

All constructs were subcloned into pMLBART or pART27 binary vector and were introduced into *Agrobacterium tumefaciens* strain GV3101 by electroporation. Transgenic lines were generated by Agrobacterium-mediated transformation, and transformants were selected on soil on the basis of resistance to the herbicide BASTA or kanamycin. Primers used to clone the different cDNAs and promoters are described in *Supplementary file 1.*

## Transcriptome analysis

RNA was extracted from tissue removed with scissors from the 0.5–1 mm marginal region of older *35S:miR319a/35S:amiR-NGA* leaves (older than that presented in *Figure 1,D1*, *Figure 1—figure supplement 3,D1* using the Qiagen RNeasy plant mini kit. cDNA was synthesized and hybridized to Affymetrix ATH1 arrays according to the manufacturer's recommendations in two biological replicates. The data have been uploaded to NCBI GEO, Series number: GSE78693 and GSE12691. Signal values were obtained and normalized using MAS5. Publicly available microarray data were obtained from GEO-OMNIBUS (GSE13596: cells isolated from various domains of the inflorescence meristem, GSE5630: dissected leaf 7 from wild-type 17-days-old plants [*Schmid et al., 2005*]), and normalized using MAS5. Digital Differentiation Index (DDI) analysis was carried out as in *Efroni et al. (2008)*, using the same set of samples for marker calibration set. Analysis was done using R 2.7.2 (www.r-project.org) and Bioconductor 2.2 (www.bioconductor.org/).

## Acknowledgements

We thank Masao Tasaka, Tom Gerats, Fred Sack, Adrienne Roeder, John Celenza, Peter Doerner, Jiří Friml, Dolf Weijers, Philip Benfey and Ji-Young Lee as well as the *Arabidopsis Biological Resource Center* (*ABRC*), Ohio State University, USA for plant material. We are grateful to David Smyth, Naomi Ori, Sureshkumar Balasubramanian, and Alexander Goldschmidt for helpful discussions as well as members of the Bowman laboratory for their valuable input. The authors acknowledge the facilities, scientific and technical assistance of Monash Micro Imaging, Monash University and Joan Clark as well as David Stewart from Zeiss for technical assistance. We also thank the *Electron Microscopy Unit*, Weizmann Institute of Science. Idan Efroni was supported by an EMBO Long term fellowship 185–2010. This work was supported by Australian Research Council grants DP110100070, DP130100177, DP160100892 (JLB) and Research Grant 863–06 from ISF (YE) and 3767–05 from BARD (YE and JLB).

## Additional information

### Funding

| Funder | Grant reference number | Author |
|---|---|---|
| Australian Research Council | DP110100070 | John Paul Alvarez<br>Chihiro Furumizu<br>John L Bowman |
| Israel Science Foundation | 863-06 | John Paul Alvarez<br>Yuval Eshed |
| United States - Israel Binational Agricultural Research and Development Fund | 3767-05 | John Paul Alvarez<br>Yuval Eshed<br>John L Bowman |
| Australian Research Council | DP130100177 | John Paul Alvarez<br>Chihiro Furumizu<br>John L Bowman |
| Australian Research Council | DP160100892 | John Paul Alvarez |
| European Molecular Biology Organization | 185-2010 | Idan Efroni |

The funders had no role in study design, data collection and interpretation, or the decision to submit the work for publication.

### Author contributions

JPA, CF, Conception and design, Acquisition of data, Analysis and interpretation of data, Drafting or revising the article; IE, Acquisition of data, Analysis and interpretation of data; YE, JLB, Conception and design, Analysis and interpretation of data, Drafting or revising the article

### Author ORCIDs

John L Bowman, http://orcid.org/0000-0001-7347-3691

## Additional files

### Supplementary files

• Supplementary file 1. Primers used for PCR-mediated cloning.

### Major datasets

The following dataset was generated:

| Author(s) | Year | Dataset title | Dataset URL | Database, license, and accessibility information |
|---|---|---|---|---|
| Alvarez JP, Furumizu C, Eshed Y, Bowman J | 2016 | Leaf margins of indeterminate Arabidopsis leaves | https://www.ncbi.nlm.nih.gov/geo/query/acc.cgi?acc=GSE78693 | Publicly available at the NCBI Gene Expression Omnibus (accession no: GSE78693). |

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

**Wolff C**. 1759. *Theoria Generationis*. Halae ad Salam, Christ. Hendel.

