## [Decision Letter]

Thank you for submitting your article "Active suppression of a leaf marginal meristem orchestrates determinate leaf growth" for consideration by *eLife*. Your article has been reviewed by three peer reviewers one of whom, Richard Amasino, is a member of our Board of Reviewing Editors, and the evaluation has been overseen by Detlef Weigel as the Senior Editor.

The reviewers have discussed the reviews with one another and the Reviewing Editor has drafted this decision to help you prepare a revised submission.

Summary:

This paper presents very interesting data that is suitable for publication in *eLife*. An understanding of how growth is limited in organs so that they achieve the proper final shape and size is important to both animal and plant developmental biologists and this work is an important advance in achieving that understanding.

The main issue to consider is whether or not this work does in fact demonstrate the existence of marginal meristems. The framing of the Introduction and other parts of the paper indicate that the data demonstrate the involvement of marginal meristems in leaf development, and that a longstanding issue is now resolved. Clearly there is proliferation at the leaf edges in the mutant-i.e., a marginal meristem-like proliferation can occur in the mutant background. But as noted below in the specific comments of reviewers, you could interpret the data as a broad failure of the mutant to exit an early leaf margin program of development and progress to the next stage of leaf blade development. Is this early leaf margin program of development really a marginal meristem? I.e., is it rigorously established that this early stage of development really has the attributes of a marginal meristem? For example, evidence that a bona fide meristem exists in the mutant situation could be that WOX genes are of central importance for the proliferation phenotype, which would be similar to what we know for root, shoot and cambium. Is the characterization of the prs wox1 double mutants sufficient to show this? There has been a Plant Cell paper on prs wox1 mutants (http://www.plantcell.org/content/24/2/519.long); the markers used in this paper are mostly different from the ones in the current paper, and in our view, the authors of the current work do not sufficiently frame their results with respect to phenotypes and conclusions from the Plant Cell paper.

We think a more balanced discussion of the marginal meristem issue would better serve the readers. Thus, in a revision, the points below ought to be addressed "head on" for the readers even if you choose to disagree with them.

1) Would it be possible to define what a marginal meristem is, in terms of where the initial (stem) cells are located? The distinction between a "marginal" and "plate" meristems has never been clear to me. The paper implies that cells at apex of the margin are not involved, unlike a shoot tip, but we can't find the evidence suggesting that they are only sub-marginal. In fact, it could be argued that loss of elongated epidermal cells from the margin is consistent with gain of meristem characters here. Similarly, might the contribution of cell layers be affected (e.g., might L1 cells divide periclinally at the margin)?

2) The evidence against marginal meristems in later stages of leaf development is convincing (e.g., from Poethig & Sussex's clonal analysis in the 1980s). However, some believe they have not been ruled out earlier in development – if you define a marginal meristem in terms of undifferentiated, dividing cells close the margin, then most of a small leaf primordium qualifies. The interpretation is therefore that reducing NGA and CIN-TCP activity reveals the existence of a marginal meristem that is normally transient, that this supports an origin of the leaf from a condensed shoot system and that gain of NGA and CIN-TCP activities might have been involved in the condensation.

Does reduction in NGA and CIN-TCP activity really reveal a marginal meristem that occurs early in development of a wild-type leaf? We agree that there are similarities in gene expression between a young wild-type primordium and the margins of an older NGA/CIN-TCP-reduced leaf, but does this also extend to cell division and cell morphology (e.g., absence of elongated marginal cells)? Do these morphologies persist towards the base of the wild-type primordium, which retains its meristem-like gene expression for longer than the tip?

3) Is there really a need to invoke an old concept (marginal meristem) that carries a lot of baggage? Especially since previous evidence indicates there is no persistent marginal meristem that establishes the leaf blade and since this paper doesn't address the existence of a marginal meristem in normal leaf growth. Indeed the continued proliferation of the leaf margins could be interpreted as an inability of the leaf margins to exit the early developmental phase that sets up the earliest outgrowth of the leaf margins and establishes the young leaf blades. Perhaps the Introduction and Conclusions could be presented in the context of either 1) addressing the problem of how organs find their correct final size or 2) what cellular components are required for leaves to progress from the pattern of growth typical of early primordium stage to the pattern of growth that defines the later stage leaf.

4) The authors find that some, but not all, of the genes involved in shoot apical meristem development also act at the proliferative leaf margins of the tcp nga mutant plants. The incomplete similarity is one of the important findings of this paper. For that reason, we suggest the following change to the Impact statement: "We describe a meristem acting at the margins of leaves, the activity of which requires some of the same, or paralogous, genetic factors as other shoot meristems, but its suppression employs factors acting primarily in leaves and other determinate organs."

5) In the absence of a clonal analysis or tracking of cells over time, which would be a big ask, marking cells with nail polish seems to provide good evidence for a marginal meristem. Marked cells appear to displaced internally, implying that growth occurs between them and the margin. Could we have this in the main figures and in more detail (e.g., showing whether some marked cells remain at the margin, consistent with sub-marginal initials and whether the growth separating the marks from the margin involves cell division).

6) The authors concentrate on development occurring at the leaf margins and this is indeed dramatic. But several of the genes targeted are expressed in leaf domains beyond the margins (e.g. TCP3,4 and NGA1,4). Do non-marginal regions of the leaf display differences in growth or are abnormalities limited to the margins? From some of the images, it appears that the central regions have relatively sparse venation – do NGA and TCP factors promote higher order venation in the central part of the leaf?

7) It would be good to establish the extent to which genes are knocked down by the artificial microRNAs or natural microRNAs in this study. Are all members of the gene family knocked down to the same extent? It should be relatively easy to do qRT-PCR experiments to answer this. Controls for more distantly related gene family members should be included.

8) The authors concentrate on development occurring at the leaf margins and this is indeed dramatic. But several of the genes targeted are expressed in leaf domains beyond the margins (e.g. TCP3,4 and NGA1,4). Do non-marginal regions of the leaf display differences in growth or are abnormalities limited to the margins? From some of the images, it appears that the central regions have relatively sparse venation – do NGA and TCP factors promote higher order venation in the central part of the leaf.

---

## [Author Response]

*1) Would it be possible to define what a marginal meristem is, in terms of where the initial (stem) cells are located? The distinction between a "marginal" and "plate" meristems has never been clear to me. The paper implies that cells at apex of the margin are not involved, unlike a shoot tip, but we can't find the evidence suggesting that they are only sub-marginal. In fact, it could be argued that loss of elongated epidermal cells from the margin is consistent with gain of meristem characters here. Similarly, might the contribution of cell layers be affected (e.g., might L1 cells divide periclinally at the margin)?*

We have extended the Introduction and Discussion sections to include more detailed descriptions of the literature. Much of point 1 is discussed in the literature of the middle of last century, whereby a marginal meristem was defined as including both epidermal cells that divide only anticlinally (an thus producing only abaxial and adaxial epidermis), and sub- epidermal (sub-marginal) cells that divide both anticlinally and periclinally (producing all internal cell layers). The marginal meristem was thought to be restricted to only a few cells at the margin. The loss of differentiated marginal epidermal cells in the NGA/TCP loss-of- function lines is consistent with this view. The plate meristem was defined as being proximal to the margin, extending over a much larger spatial area than the marginal meristem.

In our revised version we argue that leaf primordia as a whole are made of a leaf meristem. This is clearly evident in anatomical sections as presented in Figure 4. We next argue that gradual differentiation imposed by leaf specific programs acts to spatially restrict this meristem to marginal and basal domain. Thus, the marginal leaf meristem does not represent a unique identity, but rather a specific stage during leaf development that can last for different periods in different plants as extensively described by Hagemann and Gleisberg in 1996.

We also argue that the leaf meristem is maintained and defined by WOX gene expression. In the NGA/TCP loss-of-function lines, this meristem continues unabated for the life of the plant, but in wild-type plants it is shut down early. Just as in the SAM or root, loss-of-function of the WOX genes results not in a complete loss of growth, but a failure to maintain a pluripotent stem cell population, e.g. wuschel mutants produce a few leaves but the failure to maintain a central zone results in an eventual loss of production of organs from the SAM. In the leaf, loss of wox1 and prs results in an analogous phenotype — production of a bit of leaf tissue but a failure to maintain continued marginal growth. Thus, prs wox1 mutations are epistatic to the NGA/TCP loss-of-function alleles. Further evidence of the role of these marginal cells comes from an experiment whereby the dimiR-NGA/TCP was driven with the PRS regulatory sequences, which also results in indeterminate meristematic growth at the leaf margin. While a major caveat may include inherent feedbacks, this experiment does suggest that the repression of WOX gene expression at the leaf margin is critical for whole leaf differentiation. Given our knowledge of hormone production in leaf margins, particularly auxin, this observation is not surprising.

*2) The evidence against marginal meristems in later stages of leaf development is convincing (e.g., from Poethig & Sussex's clonal analysis in the 1980s). However, some believe they have not been ruled out earlier in development – if you define a marginal meristem in terms of undifferentiated, dividing cells close the margin, then most of a small leaf primordium qualifies. The interpretation is therefore that reducing NGA and CIN-TCP activity reveals the existence of a marginal meristem that is normally transient, that this supports an origin of the leaf from a condensed shoot system and that gain of NGA and CIN-TCP activities might have been involved in the condensation.*

These statements are in accordance to the literature, e.g. that nearly the entire leaf primordium at an early stage is meristematic. This idea is further supported by classical experiments on the effects of surgical leaf primordia dissection at different stages, a discussion of which can be found in the modified Discussion. This is exactly the idea we were attempting to portray in our original paper, but it was perhaps too cryptic. We do hope that the new version makes our point clearer.

*Does reduction in NGA and CIN-TCP activity really reveal a marginal meristem that occurs early in development of a wild-type leaf? We agree that there are similarities in gene expression between a young wild-type primordium and the margins of an older NGA/CIN-TCP-reduced leaf, but does this also extend to cell division and cell morphology (e.g., absence of elongated marginal cells)? Do these morphologies persist towards the base of the wild-type primordium, which retains its meristem-like gene expression for longer than the tip?*

We have included additional observations that support the idea that the leaf marginal cells in NGA/TCP loss-of-function alleles resemble at the morphological and anatomical levels that cells at the margins of wild-type leaf primordia. See Figure 1—figure supplement 5–Figure 1—figure supplement 6.

*3) Is there really a need to invoke an old concept (marginal meristem) that carries a lot of baggage? Especially since previous evidence indicates there is no persistent marginal meristem that establishes the leaf blade and since this paper doesn't address the existence of a marginal meristem in normal leaf growth. Indeed the continued proliferation of the leaf margins could be interpreted as an inability of the leaf margins to exit the early developmental phase that sets up the earliest outgrowth of the leaf margins and establishes the young leaf blades. Perhaps the Introduction and Conclusions could be presented in the context of either 1) addressing the problem of how organs find their correct final size or 2) what cellular components are required for leaves to progress from the pattern of growth typical of early primordium stage to the pattern of growth that defines the later stage leaf.*

This appears to be a matter of semantics, whether one calls the meristematic activity at the margins of early wild-type leaf primordia and which continues in NGA/TCP loss-of-function allele a marginal meristem or rather early leaf meristematic activity that continues at the margins in NGA/TCP loss-of-function alleles. We would argue this amount to the same thing, if one accepts that the early leaf primordium is meristematic, especially at its margin.

We are not sure how our work (1) addresses how organs contribute to the correct final size, as this is a combination of both meristematic activity and cell enlargement. We only wish to state that local continued proliferation can contribute to leaf form, and that the temporal and spatial shutting down on the meristematic margin could be one mechanism by which leaf shape is sculpted. We are also not sure how our work relates to the identification of (2) cellular components that are required for leaves to progress from the pattern of growth typical of early primordium stage to the pattern of growth that defines the later stage leaf.

Does this mean what causes cells to differentiate once displaced from the meristem? If so, neither NGA nor TCP are required for this as cells displaced from the meristematic margin in the NGA/TCP loss-of-function alleles gradually differentiate into regular leaf tissues.

*4) The authors find that some, but not all, of the genes involved in shoot apical meristem development also act at the proliferative leaf margins of the tcp nga mutant plants. The incomplete similarity is one of the important findings of this paper. For that reason, we suggest the following change to the Impact statement: "We describe a meristem acting at the margins of leaves, the activity of which requires some of the same, or paralogous, genetic factors as other shoot meristems, but its suppression employs factors acting primarily in leaves and other determinate organs."*

This has been altered to reflect the suggestion of the reviewers. Our idea is that, since leaves evolved from shoot systems, similar molecular mechanisms may be involved in their potentials

“they could either share genetic components, or use paralogous components, or in some cases dispense with certain factors.”

*5) In the absence of a clonal analysis or tracking of cells over time, which would be a big ask, marking cells with nail polish seems to provide good evidence for a marginal meristem. Marked cells appear to displaced internally, implying that growth occurs between them and the margin. Could we have this in the main figures and in more detail (e.g., showing whether some marked cells remain at the margin, consistent with sub-marginal initials and whether the growth separating the marks from the margin involves cell division).*

We have moved these experimental results to the main text, see Figure 1. One problem with the proposed experiments is the expectation suggested. The central zone of the shoot apical meristem has been described as a permanent office with no permanent workers (Newman 1965), and there is no reason to suspect that the meristematic region at the leaf margin is any different. Thus, even if the experiment was feasible, we might not expect any markers, e.g. carbon particles, to reside continually at the margin. A clonal analysis experiment in the NGA/TCP loss-of-function background could provide some insight, but we consider this to be beyond the scope of this work.

*6) The authors concentrate on development occurring at the leaf margins and this is indeed dramatic. But several of the genes targeted are expressed in leaf domains beyond the margins (e.g. TCP3,4 and NGA1,4). Do non-marginal regions of the leaf display differences in growth or are abnormalities limited to the margins? From some of the images, it appears that the central regions have relatively sparse venation – do NGA and TCP factors promote higher order venation in the central part of the leaf?*

While we have concentrated on the marginal aspects of growth and differentiation, the observation that both NGA and TCP are broadly expressed in developing leaves suggests a function in regions of the leaf that are undergoing differentiation. In an attempt to separate the marginal function from other potential functions, the dimiR-NGA/TCP was driven with the PRS regulatory sequences. Comparisons of sections of these leaves with those from the more constitutively expressed dimiR reveal effects on cell expansion that appear independent from the phenotype of indeterminate growth — this data has been added in Figure 1—figure supplement 13.

Concerning possible differences vasculature development and ATHB8 expression (old Figure 1 1 K, L; new Figure 1; old supplementary Figure 1 sup 7 new supplementary Figure 1 sup 10). We reevaluated vascular development using MP/ARF5:GFP as a provascular marker as shown in Figure 1—figure supplement 7. These data do not suggest a significant difference in older, more internal lamina tissues of 35S:miR319a/35S:amiR-NGA versus wild-type leaves per se. However, it does appear that in 35S:miR319a/35S:amiR-NGA leaves the midrib, which is not vascularised apart from the midvein, is considerably enlarged. This gives the impression of a larger internal region without vascular expression of either ATHB8: GUS or MP/ARF5:GFP. Also there is a tendency for the cleared, highly curled 35S:miR319a/35S:amiR-NGA leaves to tear when being flattened for examination that can open the leaf up internally and further the impression of limited vascular marker expression.

*7) It would be good to establish the extent to which genes are knocked down by the artificial microRNAs or natural microRNAs in this study. Are all members of the gene family knocked down to the same extent? It should be relatively easy to do qRT-PCR experiments to answer this. Controls for more distantly related gene family members should be included.*

Relevant data has been added as a data source file (file 4). It is also noted that the 35S:miR- NGA exhibits a phenotype indistinguishable from that of the nga1,2,3,4 quadruple mutant.

*8) The authors concentrate on development occurring at the leaf margins and this is indeed dramatic. But several of the genes targeted are expressed in leaf domains beyond the margins (e.g. TCP3,4 and NGA1,4). Do non-marginal regions of the leaf display differences in growth or are abnormalities limited to the margins? From some of the images, it appears that the central regions have relatively sparse venation – do NGA and TCP factors promote higher order venation in the central part of the leaf.*

Repeat of point 6.

Below are listed = the new/modified figures and the changes along with a rational for the changes.

Figure 1

Figure 1 I have replaced the 35S:miR319a nga1,2,3,4 leaf with one that is more dramatic and more typical.

Figure 1 has been replaced by an expansion of Figure 1—figure supplement 3 D1 and D2 in keeping with the reviewers requests.

Insertion of sections Figure 1. The reviewers requested more detail on the margins and to put the data in the context of Nakata et al. 2012 paper. In that paper they present the P4-P5 leaf primordia of the wild type formed a six-cell-layered blade consisting of the epidermal, subepidermal and two inner cell layers. So, here we present similar stages in the context of the early primordial and present evidence that this is maintained in older 35S:miR319a/35S:amiR-NGA leaf margins.

Figure 1—figure supplement 3. Ongoing marginal growth in leaves with reduced CINTCP and NGA gene activities. In A, the original 35S:miR319a nga1,2,3,4 leaf was replaced with another 35S:miR319a nga1,2,3,4 leaf and a 35S:miR319a/35S:amiR-NGA leaf with a more dramatic and typical specimen.

Figure 1—figure supplement 5. SEM margins.

What was Figure 1—figure supplement 5. In the initial version had some SEM of the margins and morphological markers of the leaves and this has been split into two figures. One with SEM and the other with markers (Figure 1—figure supplement 7). Important detail was lost in the original in our opinion and this is more interesting and demonstrative.

Figure 1—figure supplement 6. Wild-type leaf margin cells exhibit a proximal to distal cell expansion gradient lost in 35S:amiR-NGA/35S:miR319a leaves.

A new figure detailing the proximal to distal marginal-cell-differentiation gradient as requested by the reviewers.

Figure 1—figure supplement 7. Morphological and marker analyses in leaves with reduced CIN-TCP and NGA gene activities. Again what was Figure 1—figure supplement 5. But the markers alone and MP/ARF5:GFP as a provascular marker included as the reviewers requested more detail on vasculature development.

Figure 1—figure supplement 8. Reduced NGATHA and CIN-TCP activity results in a meristematic leaf margin. New supplementary figure with sections which is an extension of Figure 1. Puts sections in context and suggests that internally 35S:miR319a/35S:amiR-NGA leaves appear thicker, with larger cells.

Figure 1—figure supplement 11. PRS:GUS expression. Same as the original supplement except an older 35S:miR319a/35S:amiR-NGA leaf included (E) that used to be part of Figure 3 (originally Figure 3)

Figure 1—figure supplement 13. Reduced NGATHA and CIN-TCP activity in the PRESSED FLOWER (PRS) domain alters leaf marginal growth. This is a new supplemental figure where the miR319a, amiR-NGA and miR-319a-amiR-NGA are expressed under control of the PRS promoter. This is to respond to the reviewers request to see if we can tease out the functions of the genes at the margins versus more internal regions. The microRNAs expressed at the margins via PRS influence growth and give indeterminate leaves in PRS>>miR-319a-amiR-NGA leaves. Notably the internal tissues of PRS>>miR-319a-amiR-NGA leaves do not exhibit the expanded thickness of 35S:miR319a/35S:amiR-NGA leaves suggesting a possible role for the NGATHAs, CIN-TCPs away from the margins may be to regulate cell expansion.

Figure 1—figure supplement 15. Morphological and marker analyses in cotyledons with reduced CIN-TCP and NGA gene activities. Modification of the original Figure 1—figure supplement 11. Morphological and marker analyses in cotyledons with reduced CIN-TCP and NGA gene activities from the previous version. We have additionally inserted ATML1- H2B-mYFP (margins) and MP/ARF5:GFP (vasculature) expression to be comparable with the leaf analysis and to provide supporting evidence for a marginal meristem in the embryonic cotyledons.

Figure 3. Removed Figure 3 which was the older leaf with PRS:GUS expression as this seemed redundant with PRS:GUS expression presented in Figure 1 and Figure 1—figure supplement 11. Figure 3 was then restructured to be bigger and more logical in flow. Also Figure 3 have been rearranged vertically to maintain a consistent adaxial-facing-up for transverse sections throughout the paper.

Figure 3—figure supplement 1. Arranged D-F vertically to be consistent with the sections in the rest of the paper-adaxial facing up. Added ad-adaxial, ab-abaxial.